# A systematic review of impact of person-centred interventions for serious physical illness in terms of outcomes and costs

Kennedy Bashan Nkhoma ![ORCID],[1] Amelia Cook,[2] Alessandra Giusti ![ORCID],[2] Lindsay Farrant,[3] Ruwayda Petrus,[4] I Petersen,[5] Liz Gwyther,[3] Sridhar Venkatapuram ![ORCID],[6] Richard Harding ![ORCID] [7]

For numbered affiliations see end of article.

**Correspondence to**
Dr Kennedy Bashan Nkhoma;
kennedy.nkhoma@kcl.ac.uk

## ABSTRACT

**Background** Person-centred care (PCC) is being internationally recognised as a critical attribute of high-quality healthcare. The International Alliance of Patients Organisations defines PCC as care that is focused and organised around people, rather than disease. Focusing on delivery, we aimed to review and evaluate the evidence from interventions that aimed to deliver PCC for people with serious physical illness and identify models of PCC interventions.

**Methods** Systematic review of literature using Preferred Reporting Items for Systematic Reviews and Meta-Analyses guidelines. We searched AMED, CINAHL, Cochrane Library, Embase, Medline, PsycINFO, using the following key concepts: patient/person-centred care, family centred care, family based care, individualised care, holistic care, serious illness, chronic illness, long-term conditions from inception to April 2022. Due to heterogeneity of interventions and populations studied, narrative synthesis was conducted. Study quality was appraised using the Joanna Briggs checklist.

**Results** We screened n=6156 papers. Seventy-two papers (reporting n=55 different studies) were retained in the review. Most of these studies (n=47) were randomised controlled trials. Our search yielded two main types of interventions: (1) studies with self-management components and (2) technology-based interventions. We synthesised findings across these two models:
*Self-management component*: the interventions consisted of training of patients and/or caregivers or staff. Some studies reported that interventions had effect in reduction hospital admissions, improving quality of life and reducing costs of care.
*Technology-based interventions*: consisted of mobile phone, mobile app, tablet/computer and video. Although some interventions showed improvements for self-efficacy, hospitalisations and length of stay, quality of life did not improve across most studies.

**Discussion** PCC interventions using self-management have some effects in reducing costs of care and improving quality of life. Technology-based interventions improves self-efficacy but has no effect on quality of life. However, very few studies used self-management and technology approaches. Further work is needed to identify how self-management and technology approaches can be used to manage serious illness.

**PROSPERO registration number** CRD42018108302.

## STRENGTHS AND LIMITATIONS OF THIS STUDY

⇒ A study provides a systematic review of the evidence on the impact of person-centred interventions for serious physical illness in terms of outcomes and costs.
⇒ We used robust procedures for systematic reviewing and quality assessment of the studies included, in line with Preferred Reporting Items for Systematic Reviews and Meta-Analyses reporting guidelines.
⇒ Most of the studies identified and included were conducted in high-income countries.
⇒ We conducted a narrative synthesis due to heterogeneity of the studies included (different disease population, different outcome measures and different trial end points).
⇒ Most of the studies included did not state the theoretical framework underpinning the person-centred interventions; however, many studies that reported the theoretical framework used the University of Gothenburg Centre for person-centred care of person-centred care and were conducted in Sweden across various clinical settings.

## INTRODUCTION

WHO guidance emphasises person-centredness as a core component of healthcare professionals' skills and quality healthcare.[1] Integrated, person-centred care (PCC) is essential to achieving Universal Health Coverage (UHC).[2] [3] The core elements of PCC in health policy, medicine and nursing have been described as: patient participation and involvement, patient relationship with the healthcare professionals and context where care is delivered.[4] The International Alliance of Patients' Organisations defines PCC as 'focused and organised around people, rather than disease'.[5] PCC

views individuals, families and communities as participants in health systems responsive to their needs.[6]

PCC aims to give meaningful assessment and equal weight to a patient's subjective understanding of their illness, including their needs, concerns and expectations. This occurs, alongside a biomedical diagnosis; PCC also promote their equal participation in treatment decision-making and empowers them to take greater control of their own health and management of their condition.[7]

Our first systematic review identified and appraised the empirical evidence underpinning conceptualisations of 'person-centredness' for serious illness.[8] Serious illness, as defined in that review, includes conditions that carry a high degree of clinical uncertainty, may require high care dependency because of decreased function, but may not be advanced.[9] The review concluded that PCC (through valuing the social needs of patients, promoting quality of life and reform of health structures) will improve patients' experience of interaction with healthcare systems.[8] The review also concluded that primary data are needed that investigate the meaning and practice of PCC in a diverse diagnostic groups and settings.[8]

Re-engineering health systems to deliver PCC has particular relevance to low-income and middle-income countries (LMICs).[6 10] Serious health-related suffering places a huge burden on health systems, with the greatest burden in LMICs. Projections from WHO mortality data estimate that LMICs face the largest proportional increase, largely due to ageing (155% increase in people with serious health-related suffering in the last year of life by 2060 to 5.14 million people).[11] In such contexts, serious illness also places huge psychological, social, economic, physical and spiritual burdens on patients and (largely female) family caregivers.[12–14] It carries a high risk of mortality, negatively impacts quality of life and daily function and is burdensome in symptoms, treatments and/or caregiver stress.[15]

PCC has great potential for patients, families, staff and the healthcare system in terms of engagement, enablement, management of symptoms and reduction in re-referrals, reducing readmission, frequent visits to primary care and/or emergency visits.[16] Identification, refinement, adaptation and implementation of effective PCC interventions may thus help to achieve the WHO and UHC goals. However, no review to date has aimed to identify and synthesise the evidence for the outcomes and cost of PCC across serious physical illness. We aimed to review the evidence (in terms of outcomes and costs) for interventions that aim to deliver PCC to, or enhance person-centredness of care for, adults with serious physical illness.

## METHODS

### Design

Systematic review of peer-reviewed literature drawing on Preferred Reporting Items for Systematic Reviews and Meta-Analyses (PRISMA) guidelines, with quality appraisal using the Joanna Briggs Institute Critical Appraisal checklist, and narrative synthesis of findings. A full protocol is registered with PROSPERO, CRD42018108302.[17]

### Objectives

The objectives of this review were to (i) identify models of PCC interventions for adults with serious illness and how these were delivered; (ii) determine which outcomes have been measured as end points; (iii) appraise intervention effectiveness in terms of outcomes and costs, using narrative synthesis and (iv) evaluate the quality of the evidence.

### Search strategy

The following databases were searched in January 2020: AMED, Assian, CINAHL, Cochrane Library, Embase, Medline, PsycINFO, Scopus and Web of Science. Key journals and reference lists from included studies and relevant review articles were hand searched. We conducted a search rerun limiting it from 2020 to April 2022 (online supplemental file 1).

The search strategy (table 1) was developed in consultation with an information specialist. We used the following key concepts, drawing on our prior review of the concepts and primary data underpinning PCC[8]: person/patient-centred care, family centred care, family based care, individualised care, holistic care. Data bases were searched from inception.

Reference lists of identified papers and previous systematic reviews on PCC were hand searched.

Subject headings and word truncations were entered according to requirements of each database to map all potential keywords. Group 1 concepts were combined

**Table 1** Search strategy

| Search strategy number | Key concepts | Key words |
| --- | --- | --- |
| 1 | Patient centred Family centred Person centred Individualised Holistic | Patient-centered care or patient-centred care or client-centred care or client-centered care or client-focused care or person-centred care or person-centered care or person-focused care or family-centred care or family-focused care or family-centered care or individuali?ed or holistic care or holistic health |
| 2 | Serious illness Chronic illness Long-term illness | Chronic diseases or serious illness or chronic illness or long term conditions or long term illness |

**Table 2** Inclusion and exclusion criteria

| | Inclusion | Exclusion |
|---|---|---|
| Participants | All serious physical illness as defined by Kelly *et al* 2014; 2016: serious illness is a health condition that carries a high risk of mortality AND either negatively impacts a person's daily function or quality of life, OR excessively strains their caregivers.<br>Caregivers of patients with serious physical illness defined above.<br>Healthcare professionals (doctors, nurses, social workers, etc) caring for patients with serious physical illness. | Patients with conditions considered risk factors to develop serious illness such as hypertension. |
| Interventions | Any interventions delivered using a person-centred, or client-centred, or patient-centred, or family centred approach such as involving patients in decision-making about their care, setting goals and plans, patient being involved managing their own disease, interventions focused on the whole person, holistic approach. Interventions delivered in any format that is focused on the needs of the patients. | Any interventions delivered without patient involvement or decision making about their care or helping them take actions to support themselves. |
| Studies and comparator | Published intervention studies.<br>Written in English language only.<br>Evaluations using a comparator.<br>The comparison group should either be usual care/standard care, or a comparator intervention. | Unpublished studies, studies not written in English language, conference proceedings, conference abstracts, non-randomised trials.<br>No comparison group. |
| Outcomes | Patient and family caregiver self-report outcomes, for example:<br>▶ pain and symptom prevalence and intensity/severity, interference with daily activities, knowledge and practice of self-management, quality of life;<br>▶ psychosocial outcomes such as stress, anxiety, depression, burnout, distress;<br>▶ social, practical and spiritual; knowledge of pain and/or symptom management, quality of life, psychological outcomes (anxiety, stress, depression, distress) and caregiver motivation to provide care.<br>Formal and informal health service use.<br>Costs of services. | Outcomes not related to person-centred care (outcomes not focusing on physical, psychological social and spiritual aspects of care). |

using the 'OR' function. Likewise group 2 concepts were combined using OR function. Finally, search strategies 1 and 2 were intersected using the 'AND' function.

### Eligibility criteria
The inclusion and exclusion criteria are summarised in table 2.

### Selection of studies, data collection and management
We report the search strategy process using the PRISMA flow chart.[18] All references identified by the search strategy were exported to Endnote software and deduplicated. One reviewer (KBN) independently appraised all titles and abstracts against the inclusion and exclusion criteria. If the title and abstract was obviously irrelevant, the reference was excluded at this stage. Full-text retained references were obtained and appraised against inclusion and exclusion criteria, and if the decision was unclear this was discussed with a second reviewer (AC) and if necessary adjudicated by a third (RH).

### Data extraction
KBN and AC extracted study data using methods described in the Cochrane handbook for systematic reviews of interventional studies.[19] A standardised data extraction form was used to ensure consistency in the review.[20] KBN

extracted n=46 papers and AC extracted n=26 papers, then both authors peer-reviewed data extraction. Any queries were resolved through discussion. RH reviewed the final data extraction.

The following variables were extracted: authors, year of publication, aims and objectives, setting and country, study design, selection of participants, sample characteristics, randomisation procedures, blinding of participants and outcome assessors, assessment of outcomes and measures used, description of the intervention and comparison group, intervention delivery, sample size, data analysis, loss to follow-up, findings for outcomes and costs and study conclusions (online supplemental file 2).

### Assessment of methodological quality of the studies
We applied the Joanna Briggs Institute Critical Appraisal checklist for randomised and non-randomised trials to assess methodological quality of the studies.[21] These are summarised in online supplemental file 3. This was conducted at individual study level. AC and KBN assessed each study independently, and thereafter discussed critical appraisal. Discrepancies in the assessment of quality between AC and KBN were resolved by discussion, and RH checked the critical appraisals of the papers.

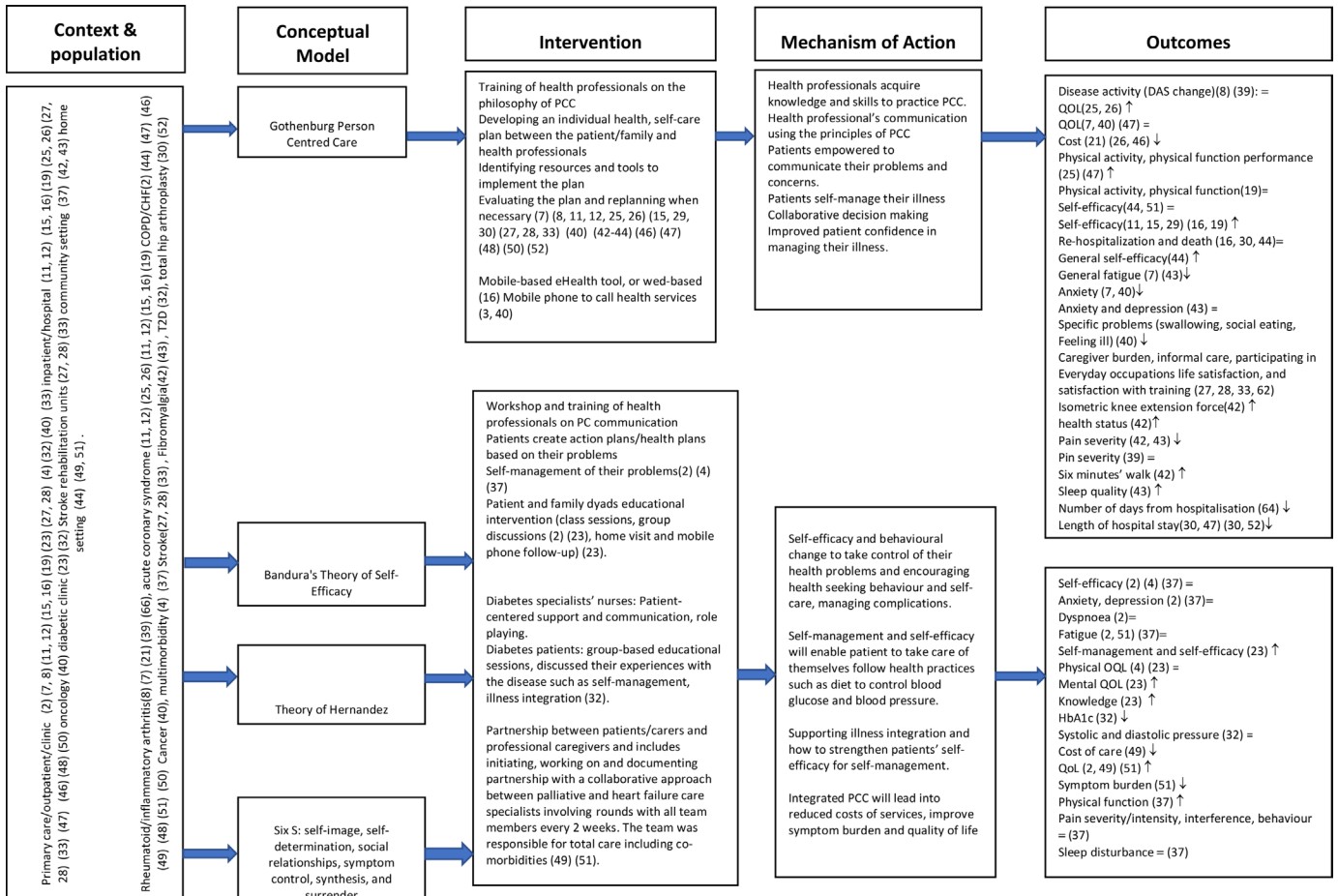

**Figure 1** Logic model for interventions with a theoretical model. HbA1c, haemoglobin A1c; PCC, person-centred care; QOL, quality of life.

## Synthesis of the evidence

Due to heterogeneity of the studies, interventions, participants and outcomes, a meta-synthesis was not conducted. We performed narrative synthesis to synthesise the findings of the different studies using the Guidance on the Conduct of Narrative Synthesis in Systematic Reviews, which consists of four elements: (1) the role of theory in evidence synthesis, (2) developing a preliminary synthesis, (3) exploring relationships within and between studies and (4) assessing the robustness of the synthesis.[22]

We developed two logic models (figures 1 and 2) to summarise the context, study population, to describe the intervention components, mechanism of action and outcomes. Figure 1 contains studies which reported a theory or conceptual framework which informed the development of the intervention. Figure 2 reports studies which did not state a theory or conceptual framework of the intervention.

A preliminary synthesis was undertaken in form of a thematic analysis involving listing and presenting results in tabular form. The results of the included studies were summarised in a narrative synthesis within a framework (participants, study aims, intervention description, usual care description, outcomes and measures used as presented in online supplemental file 2). For each study,

the effects of the intervention on the outcomes tested is provided.

We explored relationships in the data, for example, similar study design use (randomised controlled trial (RCT) vs non-RCT), similar methods of randomisation, similar intervention components and mode of delivery and similar outcomes. We then made conclusions based on the robustness of the synthesis and the quality of evidence.

### Patient and public involvement

Patient and public involvement was not conducted as part of this review.

## RESULTS

The PRISMA flow diagram (figure 3) presents the results of the search strategy. After deduplication, we screened n=5302 papers (title, abstract) and n=95 papers were retained for full-text screening. Of these, n=23 were excluded (reasons are reported in the flow chart) and n=72 papers (reporting 55 different studies) were retained in the review.

### Characteristics of the included studies

The n=56 studies included were conducted in 17 countries, the majority were high-income countries (n=13/17).

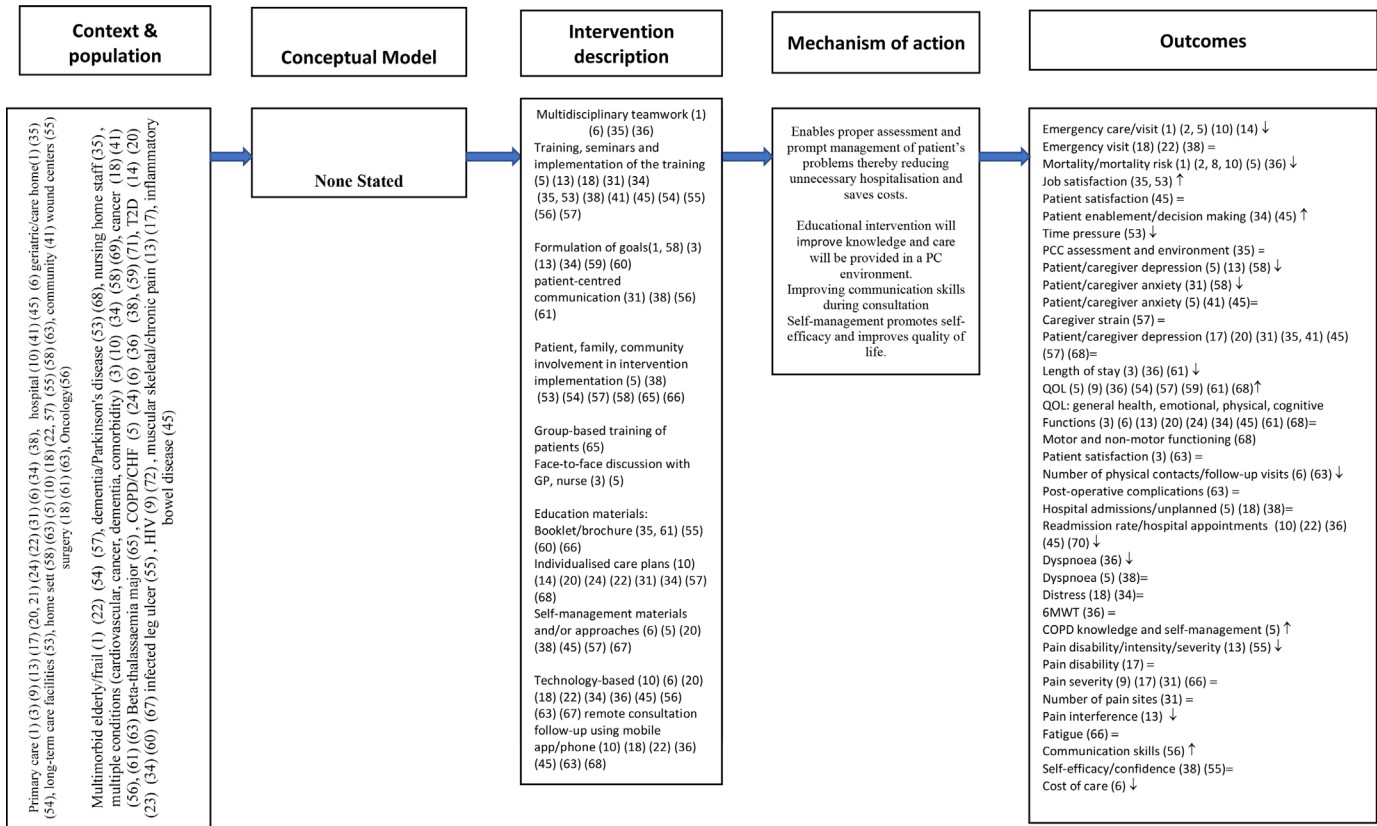

**Figure 2** Logic model for interventions without a theoretic model. 6MWT, six min walk test; CHF, chronic heart failure; COPD, chronic obstructive pulmonary disease; GP, general practitioner; PCC, person-centred care; QOL, quality of life; T2D, type 2 diabetes.

Studies were conducted predominantly in Sweden n=16, the USA n=12, Canada n=4, Germany n=4, Australia n=3, Hong Kong=3, the UK=3 and Spain n=2. One study was conducted in each of the following countries: Brasil, Denmark, Iran, Kenya, The Netherlands, New Zealand, Norway, Singapore and Thailand. A further study was

multicountry, conducted in Canada, Australia and the USA. Table 3 summarises number of studies conducted in each country.

### Study designs
Of the n=55 included studies, n=47 were RCTs, pretest and post-test experimental/controlled before and after design,[23–28] quasi-experimental study designs,[29–32] a comparative study[33] and a geographically controlled study.[34] Of the n=47 RCTs, n=11 were clustered trials.[35–44]

### Diagnostic groups
The interventions addressed the following diagnostic groups: n=12 heart failure,[24 45–55] n=9 type 2 diabetes (T2D),[34 37 38 41 56–59] n=8 chronic obstructive pulmonary disease (COPD),[43 48 60–65] n=5 cancer,[29 66–69] n=6 multimorbidity,[27 39 70–73] n=3 fibromyalgia,[42 74 75] n=3 rheumatoid arthritis,[76–78] n=2 HIV,[79 80] n=1 back pain,[81] n=1 inflammatory bowel disease (IBD),[40] n=1 osteoarthritis,[82] n=1 stroke,[83] n=1 chronic pain,[35] n=1 dementia,[84] n=1 Parkinson's disease[85] and n=1 beta-thalassaemia major.[86]

### Intervention target and delivery
The interventions were nurse-led,[33 36 61 65 69 76 80 84 87] nurse and physiotherapist-led,[48] nurse, physician and social worker-led.[27 45 71 73]

The targets of the interventions were patient and caregiver dyads[36 59 64 88] or delivered to both patients

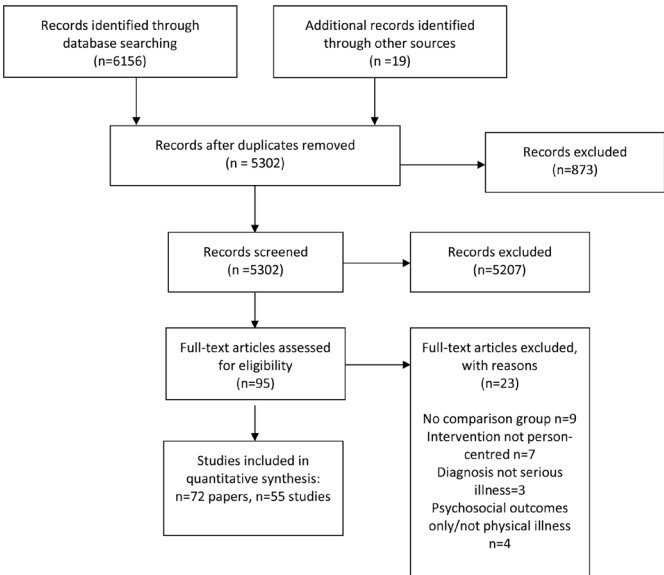

**Figure 3** Preferred Reporting Items for Systematic Reviews and Meta-Analyses flow diagram.

**Table 3** Studies and countries

| Country | Number of studies | References |
|---|---|---|
| Sweden | 16 with 31 references/papers | 23–26 28–31 45–53 56 60 66 74–78 83 89 92 95 98 99 |
| USA | 12 with 13 references/papers | 27 32 33 35–37 61 62 70 79 82 90 91 |
| Canada | 4 | 38 67 71 90 |
| Germany | 4 | 39 68 84 85 |
| Australia | 3 with 4 references/papers | 34 69 72 90 |
| Hong Kong | 3 | 57 63 88 |
| UK | 3 | 40 41 58 |
| Spain | 2 | 42 64 |
| Brasil | 1 | 81 |
| Denmark | 1 | 54 |
| Iran | 1 | 86 |
| Kenya | 1 | 80 |
| The Netherlands | 1 | 43 |
| New Zealand | 1 | 65 |
| Norway | 1 | 73 |
| Singapore | 1 | 87 |
| Thailand | 1 | 59 |
| Australia, Canada and USA | 1 | 90 |

and healthcare professionals[35 38 40 42 43 56 62 71] in T2D,[38 56] chronic pain/fibromyalgia,[35 42] COPD,[43 62] IBD[40] and multimorbidity[71] populations. The interventions were technology-based involving a tablet computer or mobile phone[37 38 40 57 63 64 69 87] or delivered to professionals such as doctors, nurses, social workers[38 41 47 50 68] working with patients with heart failure,[47 50] T2D[38 41] and cancer.[68].

### Intervention components and delivery

Interventions delivered to healthcare professionals (nurses, doctors, physiotherapists) consisted of training, mentorship and support through lecturers, seminars and/or workshops in the philosophy and delivery of PCC,[23 32 33 35 38 40 41 43 47–49 55 56 58 60 65 68 71 76 80 83 84 89 90] for example, clinical consultations using person-centred approach, person-centred communication and patient-centred self-management approach.[29 37 40 41 56 60 64 65 68 70 71] Healthcare professionals then implemented what they learnt as they provided care to the patients and/or families.

Interventions delivered to patients and/or caregivers consisted of information provision, education and training.[29–31 35 36 54 57–59 61–64 78 81 82 86 88] The interventions were either individualised and delivered face-to-face[56 62 63] or delivered in groups.[56 61 81] Educational materials, information leaflets, booklets, brochures were provided to participants.[23 29 33 41 57 58] Some interventions delivered to patients focused on developing or creating a health plan. Participants identified or set aims or goals with targets to achieve and patients identified resources and tools to achieve the targets. Healthcare professionals worked with patients to achieve the targets and care was provided in line with patient needs and wants and what matters to them.[24–28 30–32 34 35 38–40 45 47 48 55 60 62 65 66 70 72–74 77 87] The health plan was reviewed and revised when necessary.

Interventions were delivered either in nursing homes,[23] primary care/outpatient care,[35 38 39 62–64 66 71–73 76 77 80] surgical departments,[29–31 67 69] inpatient facilities[24–26 28 49 83 89 91] or in home and/or community settings.[27 32 45 46 59 61 64 70 74 79 83 85 87]

Some interventions involved using mobile technology,[27 40 59 60 62 63 66 69 87] mobile app[67] to contact patients at home. In some studies, patients in the intervention arm used either mobile-based or web-based eHealth tool pre-installed or downloaded it to use on their own mobile[53] or a tablet computer to self-monitor blood glucose and blood pressure,[57] or a web-based patient decision aid to populate their cardiometabolic and psychosocial profiles and general care priorities[38] or to complete self-assessments using a computer touch screen and to develop a self-management action plan.[37]

### Risk of bias studies included in the review

The majority of the studies (n=42) stated the method of randomisation, although this was not clearly stated in n=13 studies.[34–37 40 42 45 54 58 64 65 67 83] Twenty-eight studies achieved allocation concealment, however n=19 did not clearly state allocation concealment.[27 33 34 36 38 40 42 45 47 54 58 64 65 70 82 83 86 88 91] Blinding of participants was reported in only three studies.[35 59 72] Blinding of outcome assessors was reported in n=21 studies,[23 35 36 38 42 48 54 59 61 63 70 72 75–77 81 82 86–88 90] two studies stated that patients self-completed outcomes by post or through a web-based survey,[60 64] while n=20 studies did not clearly state if outcome assessors

were blinded. With respect to follow-up data collection, n=34 studies retained at least 80% participants to the final point of data collection. In n=19 studies, details were lacking regarding what constitutes usual care.[23 24 27 34 36 39 42 46 48 54 64 65 69 77 79 84 86–88] The following studies included all participants including those who withdraw from the study in data analysis.[40 42 47 48 59 62 63 68 71 86 92 93]

## Outcomes assessed

For patient outcomes quality of life was reported in n=22 studies.[27 34 35 37–41 43 45 46 48 49 59 61–65 71 80 85] These studies were conducted in COPD,[43 48 61–65] T2D,[34 37 38 41 59] heart failure,[45 46] chronically ill elderly,[27 39] HIV,[80] acute respiratory syndrome,[49] chronic pain,[35] Parkinson's disease,[85] IBD[40] and multimorbidity[71] populations.

General symptom burden was reported in n=4 studies in heart failure, chronically ill elderly, COPD and cancer.[27 46 62 90] Fatigue symptom was reported in n=4 studies among patients with rheumatoid arthritis, COPD, stroke, chronic illnesses (elderly populations).[48 70 77 94] Dyspnoea symptom was reported in n=3 COPD studies,[48 61 63] while only one study reported data on sleep disturbance.[70]

Pain outcomes (severity/intensity, interference and disability) were reported in nine studies,[33 35 42 70 78 80–82 95] among patients with chronic inflammatory arthritis,[78 82 95] chronic pain, low back pain, infected chronic ulcers,[35 81] HIV,[80] multiple chronic diseases[70] and fibromyalgia.[42] Nine studies reported data on communication and satisfaction with treatment.[35 39–41 48 67 78 83 95]

Self-management and related outcomes were reported in the following studies: T2D self-management,[59] COPD self-management and comorbidity,[61] enablement,[40] patient confidence in managing coronary heart disease and obtaining rheumatology care,[33 47 95] self-efficacy,[47 48 59 62 70–72] change from admission to discharge in the number of basic activities of daily living (ADLs) that the patient could perform independently,[91] performance in activities,[48 95] patient-reported health status and change in health activities[27 71] and health education impact.[72]

The main psychosocial outcomes and concerns reported were psychiatric morbidity,[80] psychological disturbance,[42 95] concerns and well-being,[41] anxiety and depression/mood,[35 37 40 48 61 70 81 90] motor function,[85] primary emotions,[40] distress[38 69–71] and decisional conflict.[38]

Caregiver outcomes assessed were depressive symptoms, caregiver strain, caregiver productivity loss,[36] caregiver quality of life[32 88] and caregiver burden.[83 88] Other caregiver outcomes were informal care that is percentage-reported providing assistance with personal ADLs,[83] participation in everyday occupations and life satisfaction.[83]

Healthcare professional outcomes included job strain,[84] transition to palliative care, general communication, involvement of significant others,[68] general practitioner (GP)'s knowledge about medication taken by the patient,[39] and intention to engage in interprofessional shared decision making.[38]

## Data on costs and healthcare utilisation

Six studies reported data on costs of healthcare utilisation,[24 45 49 64 78 91] and four on number of hospital appointments.[27 40 65 96] Two studies reported data on hospital admissions,[27 65] and three studies reported length of hospital stay.[62 91 96] Seven studies reported data on unplanned readmissions, emergency room attendance[27 34 64 69 87 93 96] and four studies reported healthcare utilisation,[27 39 72 93] and medications count (change in number of medications taken by the patient).[39]

## Data on clinical outcomes

Clinical outcomes assessed were systolic and diastolic blood pressure,[41 56 57] fasting blood sugar, haemoglobin A1c (HbA1c),[37 56–59] body mass index, haemoglobin,[41 56] lung function forced expiratory volume in 1 s/forced vital capacity ratio, exercise capacity,[63] total cholesterol to high-density lipoprotein cholesterol ratio,[37] serum ferritin, iron level, total iron binding capacity[86] and mortality.[61 63 87 93]

## Synthesis of the findings

We synthesised the findings using methods of narrative synthesis in systematic reviews.[97] A narrative synthesis is presented based on the model which informed the intervention, interventions elements/components, mechanism of action, study population, study design (RCT or non-RCT) and outcomes.

### Theoretical model/framework used by the study

The majority of the studies (n=34) did not report which theory or model informed the design or delivery of the interventions.[27 32–43 54 57 58 63–65 67–69 72 73 79 81 82 84–88 90 91] One study was informed by the theory of Hernandez,[56] three studies were developed and designed based on Bandura's self-efficacy theory[48 59 70] and another study used the person-centred palliative care model, Six S: self-image, self-determination, social relationships, symptom control, synthesis and surrender.[45 46] One study reported the chronic care model and person-centred clinical method.[71] PCC according to the University of Gothenburg Centre for Person-centred Care (GPCC) informed most of the studies conducted in Sweden.[24–26 28–31 39 47 49–51 55 60 66 74–77 83 89 98 99]

### Mechanism of action of the interventions

For the GPCC model which involved three main parameters (initiation of partnership between the patient/caregiver and healthcare professional, implementing the partnership and documenting/safeguarding the partnership). This model was applied across different settings and populations. It also involved both patients and healthcare professionals in developing and designing the intervention and implementation.

PCC requires ongoing systematic engagement between the patient and healthcare professionals. Furthermore, it requires to be adapted to each patient population (cancer, HIV, COPD, T2D, etc) and context (primary care, outpatient, residential homes, emergency care, hospital, rehabilitation, etc). Care plans, goals of care

discussed and revised as necessary continuously. Communication is also an important component in the GPCC model. Communication offered by the GPCC model gives patients (eg, inpatient setting) information and confidence about care processes and self-management of their own problems and concerns. This leads to understanding of the discharge processes and readiness and eagerness to return home which promotes self-efficacy. For the theory of Hernandez, self-efficacy and all other studies which did not state the theoretical framework, their mechanism of action were similar with the GPCC because they either had a self-management component or self-efficacy and were aimed at empowering the patient or caregiver or improving communication between the patient and the healthcare professional.

### Interventions comprising a self-management component

Fifteen RCTs consisted of a self-management intervention or component. These were conducted in COPD,[48 61 63] T2D,[56 58 59 62 64] elderly with chronic conditions,[27 70 72] cancer,[66] IBD,[40] multimorbidity[36 71] populations. All the self-management interventions were educational and consisted of training of patients and/or caregivers[27 36 48 59 63 64 66 70 72] or both healthcare professionals and patients/caregivers.[36 40 56 58] Educational sessions were either group-based[27 36 40 56 58 59] or individualised/face-to-face.[61 63] Four of the 15 studies examined effects of the intervention on hospital admissions.[27 61 63 64] Three studies showed positive benefits of self-management interventions in reducing hospital admissions. One of these four studies assessed mortality,[61] another one length of stay in the hospital[63] while one study assessed unplanned visits to the hospital.[64] All studies reported positive benefits of the intervention in reducing mortality, length of hospital stay and unplanned visits. Six of the 15 studies assessed quality of life outcomes.[36 40 59 61 63 66] In three studies, quality of life was assessed using the St George's Respiratory Questionnaire[36 61 63] and the results were significant. One study used the health-related quality of life measure and the results were non-significant, but significant on specific problems such as swallowing, social eating and feeding.[66] Three studies reported non-significant results and assessed quality of life using the IBD questionnaire,[40] the Thai Version short-form Health Survey[59] and the Chronic Respiratory Disease Questionnaire.[62] Hospital Anxiety and Depression Scale was used in three studies[40 48 61] but only one reported significant findings[61] and two reported non-significant findings.[40 48] Self-efficacy was assessed in six studies[48 59 62 70–72] with only one study reporting significant results.[59] Knowledge on self-management was reported in two studies, T2D[59] and COPD[61] populations, with both studies reporting significant differences between the intervention and control groups.[59 61]

### Technology-based interventions

Thirteen studies used technology. These were conducted among patients with T2D,[37 38 57 59] cancer,[66 67 69]

COPD,[60 63 64] chronic disease among elderly[27] and IBD.[40] Two of these studies were informed by the GPCC model[60 66] and one was informed by Bandura's model.[59] The rest were not informed by a theoretical model. Most of these technology-based intervention studies used a telephone-based intervention.[27 59 60 63 66 69] One study used a mobile app,[67] web-based,[38] four used tablet or computer technology[37 38 57 64] and three used a video.[38 40] The mechanism of action was similar across all these technology-based interventions. Patients were communicating using the phone or mobile app or tablet to ask for help if they have problems and concerns and healthcare professionals acted accordingly. This meant patient were involved in taking care of themselves and making decisions.

The outcomes however varied across these studies. Self-efficacy was examined in two studies,[59 60] with different population (COPD[60] and T2D[59]) and they used different measures to assess self-efficacy, both studies reported significant improvement in self-efficacy. Quality of life was examined in five studies[37 38 40 64 66] and they all used different measures. Only one study reported significant benefits of the intervention.[66] Hospitalisations/Rehospitalisations, length of stay, unplanned visits were reported in four studies.[27 60 63 64] All studies reported positive benefits of technology in reducing hospitalisations, length of stay and unplanned visits. Three of these studies were in COPD population,[60 63 64] one in T2D population[38] and another one study in the elderly population.[27] Two studies reported data on knowledge of management of T2D.[57 59] One study recruited participants with T2D and hypertension.[57]

However, only one study found that knowledge of T2D management was statistically significant between the intervention and control groups.[59]

One study reported data on patient assessment of chronic illness and found statistically significant differences between web-based decision aid intervention and usual care.[38]

### Synthesis based on study design

Of the n=55 included studies, n=6 studies (n=10 papers) were non-RCT.[23–26 28–32 73] Participants in these studies were elderly people with multimorbidity,[73] total hip replacement,[30 31] patients with cancer,[29] chronic heart failure,[24–26 28] patients approaching death and their family caregivers,[32] healthcare professionals in nursing homes.[23] Length of stay was assessed in heart failure, cancer and hip replacement studies and was significant in all studies.[28–31] Quality of life was assessed in three studies,[25 29 32] and two studies reported statistically significant differences between two groups,[29 32] among patients with cancer[29] and family caregivers of patients approaching death.[32]

For RCT design, n=12 studies did not clearly state the methods of randomisation. These were conducted in various populations: IBD,[40] T2D,[34 58] breast reconstruction,[67] patients with stroke and their families,[83 94 98 99] multimorbidity patients and their families,[36] heart failure/

COPD,[45 46 65] chronic pain/musculoskeletal pain/fibromyalgia.[35 42]

Quality of life was assessed in seven studies and was statistically significant in three studies,[36 45 46] but was statistically non-significant in four studies.[35 40 65 94] Pain disability, intensity and interference was assessed in the chronic pain study and showed positive benefits in all outcomes,[35] while the musculoskeletal pain (MSP)/fibromyalgia assessed pain intensity and number of tender points. Only number of tender points significantly reduced in the intervention compared with the control group.[42] Healthcare utilisation was assessed in three studies.[34 65 67] Emergency and elective admission rates significantly decreased in the intervention compared with the control group in T2D study,[34] follow-up hospital visits significantly decreased in breast reconstruction study[67] while hospital admissions were not statistically significant between two groups in COPD population.[65] Caregiver outcomes: burden, mood/anxiety,[94] depression and strain[36] were not significantly different in both studies.

Thirty-nine RCTs clearly stated randomisation methods and these recruited participants from patient, family caregivers and healthcare professionals. The main patient population were COPD (n=6),[43 48 60–63] T2D (n=6),[37 38 41 56 57 59] multiple chronic conditions and/or elderly population n=7,[55 64 69 71 75 78 92] arthritis n=4,[30 36 52 61] cancer n=3,[41 70 76] acute coronary syndrome n=6,[25 33 34 39 40 96] HIV n=2 and Parkinson's disease n=1.[66 73 88]

Quality of life, self-efficacy, health utilisation and costs of care were the main outcomes reported. Quality of life was assessed in n=16 studies, with six studies reporting statistically significant results. Quality of life was significant in a study among patients with chronic multiple conditions,[72] COPD[43 61 63] and HIV,[79 80] but was not significant in T2D population,[37 38 41 59] cancer,[66] elderly with chronic conditions,[39] acute coronary syndrome,[49 89] COPD,[62] multimorbidity[71] and patients at end of life.[90]

Self-efficacy was assessed in nine studies,[33 47 48 59 60 62 71 72 89] with only two reporting positive benefits of the intervention.[47 59] Health utilisation was reported in 10 studies.[27 39 60 61 63 69 72 79 87 91] Rehospitalisations significantly improved in COPD population and chronic multiple conditions,[27 60 63 79 87] mortality also reduced in COPD and chronic multiple conditions.[60 61 87]

Healthcare use significantly reduced among the elderly with chronic conditions,[39] length of hospital stay significantly reduced in one COPD study,[63] but was non-significant in another COPD study,[62] and among older people.[91] Hospital admission/visit to emergency was not significant in COPD and cancer population.[61 69] Healthcare use was not significant in chronic multiple conditions.[72]

## Caregiver outcomes

Quality of life among caregivers and caregiver perceived burden significantly improved among family caregivers of older people in a geriatric practice.[88] In a guided care intervention, quality of chronic Illness care, work productivity loss and absenteeism improved significantly for caregivers.[36] However, depressive symptoms and caregiver strain were not significantly changed.[36] In a cluster randomised controlled trial of a client-centred, ADL intervention for caregivers of people with stroke, caregiver burden, life satisfaction, perceived burden, mood, did not differ significantly.[83]

## Healthcare professional outcomes

A training programme among oncologists resulted in significant changes in the following behavioural domains: transition to palliative care, general communication and involving significant others.[68] A patient-centred communication intervention reported that GPs knowledge about medication taken by the patient was not significant.[39] Job strain did not differ significantly between groups even though the intervention reported greater job satisfaction. Similarly, modified task and job analysis did not differ significantly, however time pressure did decrease significantly.[84] Intention to engage in interprofessional shared decision making did not differ significantly in a Canadian trial.[38]

## Costs of care and healthcare utilisation

A person-centred integrated intervention and a technology-based intervention for patients with heart failure reduced the costs of care in the Swedish and Spanish trials, a nurse-led rheumatology clinic versus rheumatologists-led clinic, and in acute coronary syndrome,[24 45 49 64 78] however costs of services were not different among elderly admitted to a unit with acute illness.[91]

Hospital appointments decreased in the PC intervention compared with control in a multicentre cluster intervention for patients with IBD[40] likewise in an interdisciplinary collaborative practice intervention hospital visits to see the physician reduced significantly.[27] Patients in the individualised care plan intervention called out the ambulance more frequent than those who received usual care,[65] even though the intervention group had more GP visits compared with control group (15.6 vs 11.6) in 12 months and the intervention group had more hospital admissions compared with the control group the differences were not statistically significant,[65] healthcare utilisation was not significantly different between a clinician-led self-management trial and usual care.[72] A quasi-experimental design also showed no significant differences on healthcare utilisation, hospitalisation, emergency department attendance.[32]

In an integrated practice unit and modified virtual ward model in Singapore, unplanned readmissions at 30, 90 and 180 days were significantly lower in the intervention group than the control group,[87] emergency department attendance were significantly lower at 30, 90 and 180 days in the intervention.[87] Likewise an interdisciplinary, collaborative practice intervention involving a primary care physician, a nurse and a social worker for community-dwelling seniors with chronic illnesses, showed significant

changes in number of hospital admissions per patient per year, percentage of patients with one or more hospital readmissions within 60 days and mean number of visits to all physicians,[27] fewer attendances at physical, occupational or speech therapy units[39] compared with control group. However, change in percentage of patients with one or more visits to the emergency department, change in proportion of patients with one or more home care visits and change in number of patients with one or more nursing home placements and emergency visits were not significant.[27] Similarly, in a centralised, nurse-delivered telephone-based service to improve care coordination and patient-reported outcomes after surgery for colorectal cancer unplanned readmission changes in emergency visits were non-significant.[69]

Mortality was significantly reduced in the community-based integrated care for frail patients with COPD.[61] Mortality was significantly lower in an integrated practice unit and modified virtual ward model.[87] A comprehensive care programme with multidisciplinary input for patients with COPD reported reduction in mortality rates compared with usual care.[63] However, a team intervention for the multimorbid elderly reported that mortality risk at 3 and 6 months follow-up were all non-significant.[93]

A technology-based intervention of a home monitoring via mobile app on the number of in-person visits following ambulatory surgery showed that follow-up visits were significantly lower after surgery in the intervention compared with the control group,[67] number of phone calls and emails made to the healthcare in 30 days after surgery were not significant.[67] A person-centred communication intervention did not lead to change in number of medications taken by patient.[39]

In a Norwegian patient-centred team intervention number of emergency admissions, sum of emergency inpatient bed days, count of emergency re-admissions within 30 days of discharge, count of planned outpatient visits, count of emergency outpatient visits, mortality risk at 3 and 6 months follow-up were all non-significant.[93]

### Clinical outcomes

Significant improvements were seen among patients with T2D and hypertension in systolic and diastolic blood pressure,[57] likewise a patient-centred education programme among newly diagnosed patients with T2D, HbA1c was significant.[58] Fasting blood sugar, HbA1c was not statistically different between the two groups.[56 57] In a self-management trial in Sweden among patients with T2D, HbA1c was significant,[56] but not significant in a Thai trial,[59] and computer-based US trial.[37] Furthermore, cholesterol levels were not different in a computer-based trial.[37] Blood pressure (both systolic and diastolic) in a T2D trial,[41 56] and haemoglobin were not significant.[41] In a T2D UK trial body mass index was significant,[41] but was not significant in a Swedish self-management trial for patients with T2D.[56] An Iranian trial to test the effect of a holistic care programme on the reduction of iron overload in patients with beta-thalassaemia major change in

serum ferritin at 3 months (mg/L), change in iron level at 3 months (μg/dL) were significant, but change in serum ferritin 1 year and 2 years postintervention, total iron binding capacity at 3 months, haemoglobin at 3 months were not significant.[86]

## DISCUSSION

Our review found a need for data on operationalising PCC in the delivery of care for patients with serious illness. Furthermore, findings show that PCC can be provided across all settings (hospitals: inpatient, outpatient, primary care, community settings and residential homes), but majorly in primary care. PCC can be achieved by involving patients, their families and healthcare professionals. PCC can also be provided using various approaches such as self-management interventions and technology-based interventions.

Most of the studies included in the review were conducted in high-income countries predominantly in Sweden and the USA, and most of the studies using technology were conducted in high-income countries. Most participants in these studies had heart failure, T2D, COPD, cancer and arthritis. The core component of the intervention included workshop training of healthcare professionals on communication skills, training patients and families on self-assessment, identifying their problems and concerns, creating action plans based on the problems, identifying resources to self-management of the problems and evaluating the care. These components are in line with a systematic review of effective elements in a patient-centred and multimorbidity care.[100] The main outcomes reported across most studies were quality of life, healthcare utilisation and self-efficacy.

Some studies found effectiveness of PCC interventions in improving quality of life, self-efficacy, health utilisation and reducing costs of care. However, some studies reported no significant differences between PCC interventions and usual care on those outcomes.

Most studies which used person-centred self-management approaches and technology demonstrated positive benefits of the interventions in reducing hospital admissions, length of stay and unplanned visits. This finding concurs with a review of self-management interventions in respiratory and cardiovascular illness which reported that self-management support interventions reduces healthcare utilisation without compromising patient health outcomes.[101] However, self-efficacy outcomes were mostly significant in technology-based interventions, but not significant across most studies which used self-management approaches. Studies reported conflicting results on quality-of-life outcomes. Three of the six studies which used self-management approaches reported statistically significant results while only one of the six technology-based interventions reported statistically significant findings. It seems that involving a person in decision making enables them to manage their own disease through technology which

leads to reduced hospital visits and length of hospital admission. Our results concur with a previous scoping review that reported positive benefits of information and communication technology PCC interventions on five main chronic diseases (diabetes, cardiovascular, chronic respiratory, stroke and cancer).[102]

In terms of synthesis based on study design, most non-RCT reported significantly improved quality of life and reduced length of hospital stay. For RCT, of the 20 studies that reported data on quality-of-life outcomes, 9 of them reported significant results, however some of these studies did not clearly state the method of randomisation. Our findings are in line with a previous review of palliative care interventions for patients with incurable illness, which concluded that quality-of-life outcomes favoured palliative care interventions.[103]

Most of the RCTs demonstrated positive effects on the interventions in reducing re/hospitalisation, and improving health utilisation, however self-efficacy was non-significant across most RCTs.

Very few studies delivered the intervention to healthcare professionals (n=4) and caregivers (n=3). Quality of life improved and perceived burden significantly reduced in two caregiver studies. Our findings concur with a review of caregiving intervention in cancer population.[104 105]

However, psychosocial outcomes remained unchanged in our review. This is contrary to a review of multicomponent and psycho-educational interventions designed to support caregivers in their role such as training, education and skill which found positive benefits in reducing depression and burden of caregiving.[106] Our data are also at odds with findings among family caregivers in oncology populations which showed improved emotional support.[104]

Studies among healthcare professionals showed positive benefits on time pressure and communication skills, but no differences were reported on knowledge and job strain outcomes. No study reported data on implementation science outcomes among healthcare professionals. The methodological quality of these studies was poor due small sample sizes, unclear randomisation methods and allocation concealment, therefore studies that reported data on caregivers and healthcare professional outcomes are inconclusive.

Only two studies from this review demonstrated that person-centred interventions were effective in reducing pain outcomes, with five studies showing that interventions had no effect on pain and physical symptoms such as fatigue, shortness of breath in COPD and heart disease populations. However, a previous review on self-initiated interventions among patients with cancer with peripheral neuropathy showed that strategies were beneficial in reducing symptoms and concerns.[107]

Patient communication and satisfaction with PCC interventions was significant in three of the six studies that reported data on this outcome. Our findings agree with a systematic review on effectiveness of communication-related quality improvement interventions for patients with advanced and serious illness which reported significant improvements on patients' satisfaction with care.[103 108]

This review has shown that PCC interventions reduced costs of care in heart failure, COPD, acute coronary syndrome and rheumatology populations. This is in line with a meta-analysis on the economics of palliative care for adults with serious illness admitted to a facility that reported lower costs of palliative care consultations than usual care.[109] Previous studies have reported that integrated palliative care (breathless support service) reduces costs in patients with cancer and their families.[110] However, the same intervention resulted in extra mean costs of £799 in non-malignant conditions and their families,[111] therefore we can attribute the differences due to diagnosis or type of serious illness.

In our review, of the six studies that reported data on costs, five reported that PCC resulted in reduction of costs of care.[24 45 49 64 78] All these studies were conducted in primary care or home setting and two of these recruited both patients and family members as study participants.[45 64] The disease conditions were CHF,[24 45 64] acute coronary syndrome[49] and rheumatoid arthritis.[78] The majority of these studies were conducted in Sweden informed by the GPCC model of care,[24 45 49 78] while one was conducted in Spain.[64]

The intervention comprised routines for establishment of a partnership between patients and/or families and healthcare professionals (who received training on how to provide PCC, developing a health plan with the patients and/or families. The health plan also contained agreed goals,[24 45 49 64 78] these interventions were integrated in primary care. In PCC interventions informed by GPCC, healthcare professionals acquire knowledge and skills to practice PCC. Presumably this reduces hospital attendance, thereby saving time and costs travelling to the health facility. However, these are not clearly stated in the studies so we can only speculate. The only study which reported non-significant differences between the intervention and control on costs of care was among elderly people admitted to a hospital unit with acute illness.[91] This study differs from the other studies in terms of setting, and it has a heterogenous group of patients with CHF, cancer, dementia, chronic lung disease, cardiovascular disease and it is not clear which model informed the intervention.

Some studies included in this review showed significant improvements in both clinical and psychosocial outcomes, while some showed no improvements in either of them. For example, among patients with beta-thalassaemia major, significant results were reported on clinical outcomes such as serum ferritin (mg/L) and iron levels (µg/dL) including change in physical activity: 6 min walk test,[86] a technology-based trial of a person-centred tablet computer-based self-monitoring system for chronic disease (T2D and/or hypertension)[57] reported significant improvement on systolic and diastolic blood pressure but did not show significant differences on fasting blood

sugar levels and patient's knowledge of T2D and hypertension. In HIV population, a Kenyan trial showed no differences between groups on the primary outcome of pain, but showed significant differences between groups on psychiatric morbidity and quality of life[80] and another study showed no significant differences on both clinical and psychosocial outcomes in T2D population.[37]

### Strengths and limitations

It is interesting to note that the majority of the studies (n=31) achieved relative complete follow-up, that is at least 80% of the participants were followed up at trial end points. This is encouraging considering that is it challenging to follow-up participants with serious illnesses. We used robust procedures for systematic reviewing and quality assessment of the studies included, in line with PRISMA reporting guidelines, however we did not use a checklist for health economic outcome studies. We only used the Joanna Briggs Institute Critical Appraisal checklist for randomised controlled studies. Furthermore, Grading of Recommendations, Assessment, Development and Evaluations was not used for the quality of evidence for each outcome.[112] Most of the studies included did not state the theoretical framework underpinning the person-centred interventions. However, many studies that reported the theoretical framework used the GPCC and were conducted in Sweden across various clinical settings. Most of the studies identified and included were conducted in high-income countries.

Meta-analysis was not possible in this review due to heterogeneity of studies. Studies were from different patient populations, different trial designs (parallel trials or clustered trials), different sample sizes, different interventions and dimensions, different outcomes and measures used, different follow-up periods and intervals and interventions delivered in different settings. Some interventions targeted healthcare professionals and outcomes assessed among patients and healthcare professionals. Some interventions targeted patients and family dyads and captured data from both patients and their families, while some interventions targeted patients only, and family caregivers only.

Furthermore, interventions were delivered or led by different groups of professionals such as nurses, physiotherapists, physicians, social workers.

Due to nature of the interventions, it was difficult to blind study participants and those delivering the intervention, however three studies blinded study participants and two studies blinded those who delivered the intervention. It is challenging to design double-blinded or triple-blinded complex person-centred interventions. However, it is possible to blind outcome assessors. In this review, n=21 studies blinded outcomes assessors and 2 studies used postal questionnaires or web-based survey.

Some studies clearly stated the PCC model which informed the intervention while some studies did not state the PCC model. We still included studies that did not state the PCC model after critically reading through the text to understand important concepts and elements of PCC such as holistic care, coordinated physical health and supportive services, person-focused care, multidisciplinary team approach, involvement of patient and family and emphasise on person and family outcomes, respectful care and responsive to individual patient preferences, needs and values to guide all clinical decisions.[113 114] It is possible that through this process, we might have missed some papers.

### Conclusions, implications for policy, practice and research

There is some evidence that PCC interventions using self-management have some effects in reducing health utilisation, costs of care and improving quality of life.

Technology-based interventions also reduces healthcare utilisation and improves self-efficacy but appears to have less effect on quality of life. However, very few studies used self-management and technology approaches. Further work is needed to identify how self-management and technology PCC approaches can be used in serious illness across different disease conditions and settings. The majority of studies clearly defined what constituted usual care or the comparator. This shows that it is possible to design and deliver a PCC intervention in different care settings where this is currently not being practised.

PCC can be designed and evaluated using robust study designs, and can be delivered in primary, secondary and tertiary care including home settings and residential homes. Institutions should therefore consider implementing PCC interventions using locally available resources.

PCC interventions can target patients, their families or healthcare professionals. PCC research has mainly focused in high-income countries, more research needs to be done in LMICs. Further work is required to consider designing and evaluating PCC interventions at community level targeting community health workers and family members. Few studies (6/55) examined costs of PCC interventions. Health service researchers should consider incorporating costs of PCC or health economic outcomes when designing and evaluating complex PCC interventions.

**Author affiliations**
[1]Florence Nightingale Faculty of Nursing Midwifery and Palliative Care, King's College London, London, UK
[2]Cicely Saunders Institute for Palliative Care, Policy and Rehabilitation, King's College London, London, UK
[3]School of Public Health and Family Medicine, University of Cape Town Faculty of Health Sciences, Cape Town, South Africa
[4]School of Applied Human Sciences, University of KwaZulu-Natal College of Humanities, Durban, South Africa
[5]Centre for Rural Health, University of KwaZulu-Natal, Durban, South Africa
[6]Global Health Institute, King's College London, London, UK
[7]Department of Palliative Care, Policy and Rehabilitation, King's College London, London, UK

**Contributors** KBN planned, conducted searches and submitted the manuscript. KBN and AC extracted data. KBN and AC assessed quality of the included studies

and compared assessments. RH reviewed data extraction and quality appraisal. AG, RP, IP, LF, LG and SV contributed to design and interpretation. All authors approved the manuscript. KBN is the guarantor.

**Funding** This work is funded by the National Institute of Health Research (NIHR) Global Health Research Unit on Health System Strengthening in sub-Saharan Africa, King's College London (GHRU 16/136/54) using UK aid from the UK Government to support global health research.

**Competing interests** None declared.

**Patient and public involvement** Patients and/or the public were not involved in the design, or conduct, or reporting, or dissemination plans of this research.

**Patient consent for publication** Not applicable.

**Provenance and peer review** Not commissioned; externally peer reviewed.

**Data availability statement** Data are available on reasonable request. Data for this review are available on reasonable request.

**ORCID iDs**
Kennedy Bashan Nkhoma http://orcid.org/0000-0002-2991-8160
Alessandra Giusti http://orcid.org/0000-0003-2667-1665
Sridhar Venkatapuram http://orcid.org/0000-0003-3076-0783
Richard Harding http://orcid.org/0000-0001-9653-8689

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
