## [Reviewer comments · BMJ Open]

ARTICLE DETAILS

TITLE (PROVISIONAL)	A systematic review of impact of person-centred interventions for serious physical illness in terms of outcomes and costs.
AUTHORS	Bashan Nkhoma, Kennedy; Cook, Amelia; Giusti, Alessandra; Farrant, Lindsay; Petrus, Ruwayda; Petersen, I.; Gwyther, Liz; Venkatapuram, Sridhar; Harding, Richard

VERSION 1 – REVIEW

REVIEWER	Senjam, Suraj Singh All India Institute of Medical Sciences, Community Ophthalmology, Dr. Rajendra Prasad Centre for Ophthalmic Sciences
REVIEW RETURNED	28-Nov-2021

GENERAL COMMENTS	Comments The article is important to generate evidence on whether Person Centred intervention is effective in terms of improving quality of life and other clinical outcomes among people with chronic debilitating health conditions. The review is well-designed, the search methodology, results are well explained. The synthesis of the findings is well presented. However, the manuscript is quite lengthy. It needs to be shortened wherever appropriate. Some comments that authors can consider while revising it. The figure number in the text is not in order. So, PRISMA figure can be mentioned on page 12. Page 13, table 3 is quite tedious. One suggestion is that it can be appended as an annexure, similarly table 4 also. These two tables are exceptionally large for a scientific journal, it is inappropriate to include such an extra huge table within a scientific manuscript. The author needs to check how the main text of the article can be shortened. Table 3, check all “yes” score numbers. There is a mismatch. Page 124, the first para, number of studies conducted in each country can be indicated in the footnote of Table 4. It seems inappropriate to cite such a large number of references in one place. Likewise in other paras also, there are too many citations at one place. Further, in study designs, n=44 studies were RCT, but there are five citations in the round bracket. Similarly, n=12 studies were for heart failure, but in the bracket, there are 9. Likewise, there are mismatches in many places. Please check throughout the manuscript. One interesting finding is the reduction in the cost of care in the case of PCC for some serious health conditions. Usually, PCC needs a multidisciplinary approach, which means there is potential for the
---

	consumption of more resources. The author can elaborate on potential factors that lead to reduction of the cost of care in PCC than usual care in the discussion. It would be interesting to highlight the amount of cost reduction in PCC compared to other modalities being compared in the results if data are available. How much is the difference from the usual care? What are potential areas that help in cost reduction that can be discussed? Is there any difference in the components of intervention or principles among these approaches (person/family/patient/holistic/client centred)? What author would like to recommend on the name such an approach out of many terms being used? Briefly, the author can share some thoughts in the discussion. PCC, after the first mention of PCC with abbreviation, later it can be used only abbreviation. For someone with serious medical illness, using PCC intervention may not help to improve the clinical outcomes that we expected. But there could be improvement in the psycho-social outcome which is missing in this review. Check all references as per journal guidelines. Two separate references list are appended. General comments There are exceptionally extra large tables that may not be suitable for a scientific biomedical journal, it can be appended as annexure. Author can check each chapter to shorten the manuscript wherever appropriate.
--	--

REVIEWER	Gyllensten, Hanna University of Gothenburg, Institute of Health and Care Sciences
REVIEW RETURNED	14-Feb-2022

GENERAL COMMENTS	The study provides an overview of costs and health outcomes measured in studies comparing person-centred care to usual care, among patients with serious physical illness. A systematic review of this subject is long overdue, and the study thus provides a useful addition to researchers of effectiveness and cost-effectiveness of person-centred care. However, some improvements to its presentation are suggested below. General comments: I struggled a bit to understand the role of the two categories of studies being presented – the self-management studies vs the technology-based studies – since it sometimes (eg in the abstract) sounded as if all studies fell into one or the other, and sometimes (eg in the results) was clear that these categories only covered some of the included papers. In page 6, rows 38-39, it sounds as if there should be other categories also, but that these two specifically needs more work. How should I understand these categories compared to other included papers, are they covering all studies or not and are they mutually exclusive? Reading your inclusion/exclusion criteria, I find the inclusion criteria “Interventions” (table 2) very broad. How do you think about person-centred care, patient-centred care, and integrative care in this? I get the feeling that you could even have expressed this as patient-involvement in healthcare instead. How do you mean that all the included papers relate to person-centred care in particular, or do you
--

not distinguish between eg person-centred and patient-centred?

It is a bit unclear to me how the logic model was developed and how I should understand this compared to the previously published logic model for person-centred care (Tabbush et al 2016)? Should I read the different boxes in the "Conceptual model" in figure 2 as if the Gothenburg model do not in any way relate to the other models reported, while the other three (Bandura, Hernandez and Six S) are interchangeable? And in figure 3, how can a reader know that this is person-centred care if the included papers listed in this figure do not report any type of person-centred care model? I guess this question also relates back to the previous comment about what is included as person-centred care, which also affects how we as readers can understand the logic model presented?

With regards to the systematic nature of the review: Have you considered doing independent screening of at least part of the hits from the initial search and part of the full text references screened in, as a quality check? Did you consider using any quality appraisal checklist for health economic studies?

With regards to reporting of results I would suggest to at least somewhere (eg page 123?) list how many studies/projects the results really includes. Now you are clarifying in table 4 that in some instances, several papers come from the same research project, but in the main text you appear to only talk about papers as isolated studies. However, that makes it sound as if there has been a very large number of research projects studying person-centred care globally, and in particular in Sweden, but that is a bit of an overestimation of the actual knowledge available. The number of papers in these cases giving more of an indication of the large body of data collected and necessary to say something about the results of these complex interventions, and also of the situation relevant for many researchers who need to present results before the whole project is completed eg to ensure future funding (thus resulting in separate papers for 6-month follow-up, 12-month follow-up, 24 month follow-up, and separate papers when retrospective register data is available et cetera).

I think the comparator should warrant some discussion in your review, in particular since you in the end draw the conclusion that institutions should consider implementing person-centred care based on these findings. I would also suggest to clarify in the conclusions that there is also a need for further studies in HIC (although less urgent than in the other country categories) and preferably also say something about the types of cost components that we currently know anything about.

You present some limitations of the included studies, but I fail to see that you have written a section on methodological considerations for your own study. Is that correct?

Minor adjustments/corrections:

Please clarify if this systematic review is an amendment/addition to your initial PROSPERO registration, because I read that one as only focusing costs and health outcomes in palliative care studies.

Page 4, row 31: I suppose it should be the Gothenburg theory, or model, of person-centred care, and not Goldenberg? Overall, I would suggest deciding on a phrase that is used throughout when

indicating this model, as it is now indicated sometimes as the Gothenburg theory, sometimes as the Gothenburg person-centred care and sometimes as a framework. In the literature it is often referred to as the “Gothenburg model”, eg the Gothenburg model for person-centred care or indicated as person-centred care as it is operationalized by the University of Gothenburg Centre for Person-centred Care. Moreover, the research centre itself should be referred to as the “University of Gothenburg Centre for Person-centred Care” (or GPCC in short), but not as the “Gothenburg Centre for Person-centred Care” (which is used in page 128).

Page 6, rows 8-9: This bullet point is now quite vague, until the main text is read. Could it be indicated even here why it is inconclusive?

Page 13 (page 14 in my printout), row 7: check sentence “...developed two a logic...”

In page 13 (rows 30-36) you write about explored relationships, but it is not clear to me how or why these specific ways of looking at the data were chosen?

I find it challenging to know how to find different references (n particular the ones included in the results/review). In table 3 is only presented first author and year. In the text you use reference numbers, and in table 4 you use first author and year plus reference number, but there are two different reference lists, and I still am unsure how to know which to look in.

In table 3 reference 15 (Fors et al 2018) you have written that blinding of outcomes assessor is N/A because it was collected by post from the patients themselves, while no other papers had that response. Why is that paper singled out? I believe many of the papers from GPCC used the same method of collecting data as that one did, so it is unclear why the response here differs, but since I could not identify which paper you referred to in each row of this table, I was not able to see if there were even other papers from this particular project. Moreover, in the discussion (page 139) you write that only 17 papers blinded outcomes assessors, but there are also several papers for which you have decided that this was not relevant. I assume that by outcomes assessor you here refer to the clinicians doing assessments (eg symptoms scales etc), but that when it's a patient assessing their own outcome you believe it is no longer relevant to be blinded, do I understand you correctly?

In table 4 I would suggest making it clear that papers referred to as 11a and 11b comes from the same project as papers 20a-e.

I would suggest writing “Quality of reporting” instead of “Methodological quality of studies” as sub-heading since this at least to some extent can be more a question of reporting than underlying methodology.

Page 127, row 35: “Six studies reported data on costs...” but did all these papers report cost comparisons or was it sometimes reported only on a descriptive level? Moreover, you sometimes in the paper refer to resource use as “health utilization”, but not here. I would suggest using either “healthcare use/utilization” or “resource use/utilization”, because I think the term “health utilization” is a bit counterintuitive (it sounds to me as if it is the health that is used, thus potentially creating new resources?).

	Page 131, row 1: If all studies reported positive benefits from technology, do you have an idea if this was caused by the person-centredness or by adding the application itself, regardless how person-centred or not? Page 131, rows 33, 35, 50: Check how you have written about statistical significance here (and elsewhere), it is now written as if an outcome measure itself can be “significant”, not as if the difference between two groups was tested and found statistically significant on some prespecified level for the p-value. Page 131, row 52: Is “unchanged” an indication of equivalence or non-inferiority, or just not “reached” statistically significant difference, maybe because of not having enough power? How should I read the first paragraph of the discussion? I was expecting an overview of key findings, but this reads more as a brief overview of the background. I could not see from where in the results section these items came, or if all came from results or if it was opinions (“crucial mechanisms”)? Page 136, rows 41-42: Check “self- and self-efficacy.” Page 137, rows 8-12: This finding is a bit difficult for me, since as far as I can see it only includes 4 papers of one type and 2 papers of the other, and thus the difference in numbers of papers showing statistically different results is 1 vs 2 papers, or do I read this incorrectly? Could there be reporting bias going on here as well, with positive results being more likely to be reported? Page 137, row 29: In which way is your result “at odds” with the previous review? Page 137, row 44: Do I understand it correctly that the non-serious illness indicated is intellectual disability, or do I look in the wrong reference list? Which were “the conflicting results”? Page 137, row 48: Why were the list/review of patient-centred frameworks relevant here? Is there overlap between the models/frameworks or included papers? Page 137, rows 54-55: How do you mean “study participants” here? Do you refer to as study “objects”, as in asked to provide informed consent? Page 140, row 1: I would suggest to change “but have no effect on” to something less strong, such as “but appears to have less effect on”.
--	---

VERSION 1 – AUTHOR RESPONSE

Reviewer: 1
Dr. Suraj Singh Senjam, All India Institute of Medical Sciences

Comments

The article is important to generate evidence on whether Person Centred intervention is effective in

terms of improving quality of life and other clinical outcomes among people with chronic debilitating health conditions.

The review is well-designed, the search methodology, results are well explained. The synthesis of the findings is well presented. **However, the manuscript is quite lengthy. It needs to be shortened wherever appropriate. Some comments that authors can consider while revising it.**

Thank you for this feedback, we have cutback text in the manuscript. For example, table 3 and 4 are now in the appendix.

The figure number in the text is not in order. So, PRISMA figure can be mentioned on page 12.

We have rearranged the order of the figures as follows:

Initially we said figure 2 and 3 representing synthesis and figure 1 PRISMA flow diagram

We now say:

Figure 1 contains studies which reported a theory or conceptual framework which informed the intervention. Figure 2 reports studies which did not state a theory or conceptual framework of the intervention. Figure 3 shows PRISMA diagram.

Page 13, table 3 is quite tedious. One suggestion is that it can be appended as an annexure, similarly table 4 also. These two tables are exceptionally large for a scientific journal, it is inappropriate to include such an extra huge table within a scientific manuscript. The author needs to check how the main text of the article can be shortened.

Thanks for this suggestion, We have put tables 3 and 4 in the appendix.

Table 3, check all "yes" score numbers. There is a mismatch.

We have checked and edited all the 'Yes' responses, there was a mismatch on Bergsten et al 2019 and Ohlen et al (2019), these have been corrected.

Page 124, the first para, number of studies conducted in each country can be indicated in the footnote of Table 4. It seems inappropriate to cite such a large number of references in one place. Likewise in other paras also, there are too many citations at one place.

Further, in study designs, n=44 studies were RCT, but there are five citations in the round bracket. Similarly, n=12 studies were for heart failure, but in the bracket, there are 9. Likewise, there are mismatches in many places. Please check throughout the manuscript.

Thanks for this observation. There were texts mismatch in some citations because some studies had more than one publication or references. In light of this we have removed all the citations in the text where we have studies with more than one reference. This was common in studies conducted in Sweden because they published several papers from the same study/population. The text and the table below highlights the changes we have made in the manuscript:

The n=57 studies included were conducted in 17 countries, the majority were high-income countries (n=13/17). Studies were conducted predominantly in Sweden n=16, USA n=12, Canada n=4, Germany n=4, Australia n=3, Hong Kong=3, UK=3 and Spain n=2. One study was conducted in each of the following countries: Brasil, Denmark, Iran, Kenya, Netherlands New Zealand, Norway, Singapore, and Thailand. A further study was multi-country, conducted in Canada, Australia, and USA.

Studies and countries

Country	Number of studies	References
----------------	--------------------------	-------------------

Sweden	16 with 31 references/papers	(1-31)
USA	12 with 13 references/papers	(32-44),
Canada	4 with 5 references/papers	(35, 45-48)
Germany	4	(49-52)
Australia	3 with 4 references/papers	(35, 53-55)
Hong Kong	3	(56-58)
UK	3	(59-61)
Spain	2	(62, 63)
Brasil	1	(64)
Denmark	1	(65)
Iran	1	(66)
Kenya	1	(67)
Netherlands	1	(68)
New Zealand	1	(69)
Norway	1	(70)
Singapore	1	(71)
Thailand	1	(72)
Australia, Canada, and USA	1	(35)

One interesting finding is the reduction in the cost of care in the case of PCC for some serious health conditions. Usually, PCC needs a multidisciplinary approach, which means there is potential for the consumption of more resources. The author can elaborate on potential factors that lead to reduction of the cost of care in PCC than usual care in the discussion.

It would be interesting to highlight the amount of cost reduction in PCC compared to other modalities being compared in the results if data are available. How much is the difference from the usual care? What are potential areas that help in cost reduction that can be discussed?

Is there any difference in the components of intervention or principles among these approaches (person/family/patient/holistic/client centred)? What author would like to recommend on the name such an approach out of many terms being used? Briefly, the author can share some thoughts in the discussion.

Thanks for this observation, it is difficult to discuss what factors resulted in reduction in costs of care in our review. The reviewer is right that some interventions will lead into high costs of care while some reduce costs of care. This varies in different settings, population, intervention content etc. We have added the following text to the discussion:

Previous studies have reported that integrated palliative care (breathless support service) reduces costs in cancer patients and their families (73). However the same intervention resulted in extra mean costs of £799 in non-malignant conditions and their families (74), therefore we can attribute the differences due to diagnosis or type of serious illness.

In this review, of the six studies that reported data on costs, five reported that PCC resulted in reduction of costs of care (1, 12, 24, 30, 63). All these studies were conducted in primary care or home setting and two of these recruited both patients and family members as study participants (1,

63). The disease conditions were CHF(1, 24, 63), acute coronary syndrome (12) and rheumatoid arthritis (30). These studies were conducted in Sweden informed by the GPCC model of care (1, 12, 24, 30), and Spain (63).

The intervention comprised of routines for establishment of a partnership between patients, and/or families and healthcare professionals (who received training on how to provide person-centred care, developing a health plan with the patients and/or families. The health plan also contained agreed goals (1, 12, 24, 30, 63), These interventions were integrated in primary care. In person-centred care interventions informed by GPCC, healthcare professionals acquire knowledge and skills to practice PCC. Presumably this reduces hospital attendance thereby saving time and costs travelling to the health facility. However, these are not clearly stated in the studies so we can only speculate. The only study which reported nonsignificant differences between the intervention and control on costs of care was among elderly people admitted to a hospital unit with acute illness(34). This study differs from the other studies in terms of setting, and it has a heterogenous group of patients with CHF, cancer, dementia, chronic lung disease, cardiovascular disease and it is not clear which model informed the intervention.

PCC, after the first mention of PCC with abbreviation, later it can be used only abbreviation.

Thanks, we have checked through the manuscript, and we use PCC.

For someone with serious medical illness, using PCC intervention may not help to improve the clinical outcomes that we expected. But there could be improvement in the psycho-social outcome which is missing in this review.

Thanks for this comment. The reviewer is right some person-centred care interventions were designed to improve clinical outcomes, such as pain, physical symptoms but did not bring this expected change but changes were seen on psychosocial outcomes. In some studies, no changes were reported in any of these outcomes. In light of this we have added the following text in the discussion section:

Some studies included in this review showed significant improvements in both clinical, and psychosocial outcomes, while some showed no improvements in either of them. For example among beta-thalassaemia major patients, significant results were ported on clinical outcomes such as serum ferritin (mg/L) and iron levels (micrograms/dL) including change in physical activity: six-minute walk test (6MWT) (66), a technology-based trial of a person-centred tablet computer-based self-monitoring system for chronic disease (T2D and/or hypertension)(56) reported significant improvement on systolic and diastolic blood pressure but did not show significant differences on fasting blood sugar levels and patient's knowledge of T2D and hypertension. In HIV population a Kenyan trial showed no differences between groups on the primary outcome of pain, but showed significant differences between groups on psychiatric morbidity and quality of life (67) and another study showed no significant differences on both clinical and psychosocial outcomes in T2D population (41).

Check all references as per journal guidelines. Two separate references list are appended.

Thanks for this observation, there are references in each of the documents/files attached separately (cover letter, figures, tables) because we have cited these references within the text, however the main manuscript document contains all the references, and these are the ones to be used when considering this manuscript for publication.

General comments

There are exceptionally extra large tables that may not be suitable for a scientific biomedical journal, it can be appended as annexure. Author can check each chapter to shorten the manuscript wherever appropriate.

Thanks for this observation, we have removed a table with study findings and critical appraisal from the main text, we have attached these to the appendix.

References

1. Sahlen K-G, Boman K, Brannstrom M. A cost-effectiveness study of person-centered integrated heart failure and palliative home care: Based on a randomized controlled trial. *Palliative Medicine*. 2016;30(3):296-302.
2. Brannstrom M, Boman K. Effects of person-centred and integrated chronic heart failure and palliative home care. *PREFER: a randomized controlled study*. *European Journal of Heart Failure*. 2014;16(10):1142-51.
3. Fors A, Swedberg K, Ulin K, Wolf A, Ekman I. Effects of person-centred care after an event of acute coronary syndrome: Two-year follow-up of a randomised controlled trial. *International Journal of Cardiology*. 2017;249:42-7.
4. Jutterstrom L, Hornsten A, Sandstrom H, Stenlund H, Isaksson U. Nurse-led patient-centered self-management support improves HbA1c in patients with type 2 diabetes-A randomized study. *Patient Education and Counseling*. 2016;99(11):1821-9.
5. Bertilsson AS, Eriksson G, Ekstam L, Tham K, Andersson M, von Koch L, et al. A cluster randomized controlled trial of a client-centred, activities of daily living intervention for people with stroke: one year follow-up of caregivers. *Clinical rehabilitation*. 2016;30(8):765-75.
6. Zakrisson AB, Arne M, Hasselgren M, Lisspers K, Ställberg B, Theander K. A complex intervention of self-management for patients with COPD or CHF in primary care improved performance and satisfaction with regard to own selected activities; A longitudinal follow-up. *J Adv Nurs*. 2019;75(1):175-86.
7. Larsson I, Bergman S, Bremander A. Person-centred care (PCC) may improve health care consumer skills more than regular care-an RCT in patients with CIA undergoing biological therapy. *Annals of the Rheumatic Diseases*. 2015;2:104.
8. Bergsten U, Almedhed K, Baigi A, Jacobsson LTH. A randomized study comparing regular care with a nurse-led clinic based on tight disease activity control and person-centred care in patients with rheumatoid arthritis with moderate/high disease activity: A 6-month evaluation. *Musculoskeletal Care*. 2019;17(3):215-25.
9. Böckberg C, Behm L, Wallerstedt B, Ahlström G. Evaluation of person-centeredness in nursing homes after a palliative care intervention: pre- and post-test experimental design. *BMC Palliative Care*. 2019;18(1):44.
10. Öhlén J, Sawatzky R, Pettersson M, Sarenmalm EK, Larsdotter C, Smith F, et al. Preparedness for colorectal cancer surgery and recovery through a person-centred information and communication intervention - A quasi-experimental longitudinal design. *PLoS One*. 2019;14(12):e0225816.
11. Pirhonen L, Olofsson EH, Fors A, Ekman I, Bolin K. Effects of person-centred care on health outcomes-A randomized controlled trial in patients with acute coronary syndrome. *Health Policy*. 2017;121(2):169-79.
12. Pirhonen L, Bolin K, Olofsson EH, Fors A, Ekman I, Swedberg K, et al. Person-Centred Care in Patients with Acute Coronary Syndrome: Cost-Effectiveness Analysis Alongside a Randomised Controlled Trial. *PharmacoEconomics - Open*. 2019;3(4):495-504.
13. Fors A, Blanck E, Ali L, Swedberg K, Ekman I. Person-centred telephone-support is effective in patients with chronic obstructive pulmonary disease and/or chronic heart failure-six-month follow-up of a randomized controlled trial. *European Journal of Heart Failure*. 2018;20 (Supplement 1):194.
14. Feldthusen C, Dean E, Forsblad-d'Elia H, Mannerkorpi K. Effects of Person-Centered Physical Therapy on Fatigue-Related Variables in Persons With Rheumatoid Arthritis: A Randomized Controlled Trial. *Arch Phys Med Rehabil*. 2016;97(1):26-36.
15. Fors A, Taft C, Ulin K, Ekman I. Person-centred care improves self-efficacy to control symptoms after acute coronary syndrome: A randomized controlled trial. *European Journal of Cardiovascular Nursing*. 2016;15(2):186-94.
16. Fors A, Gyllensten H, Swedberg K, Ekman I. Effectiveness of person-centred care after acute coronary syndrome in relation to educational level: Subgroup analysis of a two-armed randomised controlled trial. *Int J Cardiol*. 2016;221:957-62 PubMed .
17. Fors A, Ekman I, Taft C, Björkelund C, Frid K, Larsson ME, et al. Person-centred care after acute coronary syndrome, from hospital to primary care - A randomised controlled trial. *Int J Cardiol*. 2015;187:693-9 PubMed .
18. Wolf A, Fors A, Ulin K, Thorn J, Swedberg K, Ekman I. An eHealth Diary and Symptom-Tracking Tool Combined With Person-Centered Care for Improving Self-Efficacy After a Diagnosis of Acute Coronary Syndrome: A Substudy of a Randomized Controlled Trial. *J Med Internet Res*. 2016;18(2): PubMed e40.

19. Hansson E, Carlström E, Olsson LE, Nyman J, Koinberg I. Can a person-centred-care intervention improve health-related quality of life in patients with head and neck cancer? A randomized, controlled study. *BMC Nurs.* 2017;16:9.
20. Bertilsson A-S, Ranner M, von Koch L, Eriksson G, Johansson U, Ytterberg C, et al. A client-centred ADL intervention: three-month follow-up of a randomized controlled trial. *Scandinavian journal of occupational therapy.* 2014;21(5):377-91.
21. Guidetti S, Ranner M, Tham K, Andersson M, Ytterberg C, von Koch L. A "Client-Centred Activities of Daily Living" Intervention for Persons with Stroke: One-Year Follow-up of a Randomized Controlled Trial. *Journal of Rehabilitation Medicine.* 2015;47(7):605-11.
22. Larsson A, Palstam A, Löfgren M, Ernberg M, Bjersing J, Bileviciute-Ljungar I, et al. Resistance exercise improves muscle strength, health status and pain intensity in fibromyalgia--a randomized controlled trial. *Arthritis Res Ther.* 2015;17(1):161 PubMed .
23. Ericsson A, Palstam A, Larsson A, Löfgren M, Bileviciute-Ljungar I, Bjersing J, et al. Resistance exercise improves physical fatigue in women with fibromyalgia: a randomized controlled trial. *Arthritis Res Ther.* 2016;18:176.
24. Hansson E, Ekman I, Swedberg K, Wolf A, Dudas K, Ehlers L, et al. Person-centred care for patients with chronic heart failure - a cost-utility analysis. *Eur J Cardiovasc Nurs.* 2016;15(4):276-84.
25. Ulin K, Olsson LE, Wolf A, Ekman I. Person-centred care - An approach that improves the discharge process. *European Journal of Cardiovascular Nursing.* 2016;15(3):e19-26.
26. Ekman I, Wolf A, Olsson LE, Taft C, Dudas K, Schaufelberger M, et al. Effects of person-centred care in patients with chronic heart failure: the PCC-HF study. *European Heart Journal.* 2012;33(9):1112-9.
27. Dudas K, Olsson LE, Wolf A, Swedberg K, Taft C, Schaufelberger M, et al. Uncertainty in illness among patients with chronic heart failure is less in person-centred care than in usual care. *European Journal of Cardiovascular Nursing.* 2013;12(6):521-8.
28. Olsson L-E, Karlsson J, Berg U, Kärrholm J, Hansson E. Person-centred care compared with standardized care for patients undergoing total hip arthroplasty—a quasi-experimental study. *Journal of Orthopaedic Surgery and Research.* 2014;9(1):95.
29. Olsson LE, Hansson E, Ekman I. Evaluation of person-centred care after hip replacement-a controlled before and after study on the effects of fear of movement and self-efficacy compared to standard care. *BMC Nurs.* 2016;15(1):53.
30. Larsson I, Fridlund B, Arvidsson B, Teleman A, Svedberg P, Bergman S. A nurse-led rheumatology clinic versus rheumatologist-led clinic in monitoring of patients with chronic inflammatory arthritis undergoing biological therapy: A cost comparison study in a randomised controlled trial. *BMC Musculoskeletal Disorders.* 2015;16 (1) (no pagination)(817).
31. Larsson I, Fridlund B, Arvidsson B, Teleman A, Bergman S. Treatment outcomes from a nurse-led rheumatology clinic in monitoring of anti-TNF therapy-a randomised controlled trial. *Arthritis and Rheumatism.* 2012;10):S667.
32. Kelechi TJ, Mueller M, Spencer C, Rinard B, Loftis G. The effect of a nurse-directed intervention to reduce pain and improve behavioral and physical outcomes in patients with critically colonized/infected chronic leg ulcers. *Journal of Wound, Ostomy, & Continence Nursing.* 2014;41(2):111-21.
33. Sommers LS, Marton KI, Barbaccia JC, Randolph J. Physician, nurse, and social worker collaboration in primary care for chronically ill seniors. *Archives of Internal Medicine.* 2000;160(12):1825-33.
34. Landefeld CS, Palmer RM, Kresevic DM, Fortinsky RH, Kowal J. A randomized trial of care in a hospital medical unit especially designed to improve the functional outcomes of acutely ill older patients. *New England Journal of Medicine.* 1995;332(20):1338-44.
35. Chochinov HM, Kristjanson LJ, Breitbart W, McClement S, Hack TF, Hassard T, et al. Effect of dignity therapy on distress and end-of-life experience in terminally ill patients: a randomised controlled trial. *Lancet Oncology.* 2011;12(8):753-62.
36. Dobscha SK, Corson K, Perrin NA, Hanson GC, Leibowitz RQ, Doak MN, et al. Collaborative care for chronic pain in primary care: a cluster randomized trial. *JAMA.* 2009;301(12):1242-52.
37. Hernandez C, Alonso A, Garcia-Aymerich J, Serra I, Marti D, Rodriguez-Roisin R, et al. Effectiveness of community-based integrated care in frail COPD patients: A randomised controlled trial. *npj Primary Care Respiratory Medicine.* 2015;25 (no pagination)(15022).
38. Wolff JL, Giovannetti ER, Boyd CM, Reider L, Palmer S, Scharfstein D, et al. Effects of guided care on family caregivers. *The Gerontologist.* 2010;50(4):459-70.

39. Murphy SL, Lyden AK, Smith DM, Dong Q, Koliba JF. Effects of a tailored activity pacing intervention on pain and fatigue for adults with osteoarthritis. *American Journal of Occupational Therapy*. 2010;64(6):869-76.
40. Britt HR, JaKa, M. M., Fernstrom, K. M., Bingham, P. E., Betzner, A. E., Taghon, J. R., Shippee, N. D., Shippee, T. P., Schellinger, S. E., & Anderson, E. W. . Quasi-Experimental Evaluation of LifeCourse on Utilization and Patient and Caregiver Quality of Life and Experience. . *The American journal of hospice & palliative care* 2019;36(5):408-16.
41. Glasgow RE, Nutting PA, King DK, Nelson CC, Cutter G, Gaglio B, et al. Randomized effectiveness trial of a computer-assisted intervention to improve diabetes care. *Diabetes Care*. 2005;28(1):33-9.
42. Gustafson D, Hawkins R, Boberg E, Bricker E, Pingree S, Chan C-L. The use and impact of a computer-based support system for people living with AIDS and HIV infection . *Annual Symposium on Computer Application [sic] in Medical Care Symposium on Computer Applications in Medical Care*. 1994;1(2):604-8.
43. Mielenz TJ, Tracy M, Jia H, Durbin LL, Allegrante JP, Arniella G, et al. Creation of the Person-Centered Wellness Home in Older Adults. *Innovation in Aging*. 2020;4(1).
44. Thom DH, Willard-Grace R, Tsao S, Hessler D, Huang B, DeVore D, et al. Randomized Controlled Trial of Health Coaching for Vulnerable Patients with Chronic Obstructive Pulmonary Disease. *Annals of the American Thoracic Society*. 2018;15(10):1159-68.
45. Armstrong KA, Coyte PC, Brown M, Beber B, Semple JL. Effect of home monitoring via mobile app on the number of in-person visits following ambulatory surgery a randomized clinical trial. *JAMA Surgery*. 2017;152(7):622-7.
46. Yu C, Choi D, Bruno BA, Thorpe KE, Straus SE, Cantarutti P, et al. Impact of MyDiabetesPlan, a Web-Based Patient Decision Aid on Decisional Conflict, Diabetes Distress, Quality of Life, and Chronic Illness Care in Patients With Diabetes: Cluster Randomized Controlled Trial. *J Med Internet Res*. 2020;22(9):e16984.
47. Fortin M, Stewart M, Ngangue P, Almirall J, Bélanger M, Brown JB, et al. Scaling Up Patient-Centered Interdisciplinary Care for Multimorbidity: A Pragmatic Mixed-Methods Randomized Controlled Trial. *Annals of Family Medicine*. 2021;19(2):126-34.
48. Yu C, Choi D, Bruno BA, Thorpe KE, Straus SE, Cantarutti P, et al. Impact of MyDiabetesPlan, a Web-Based Patient Decision Aid on Decisional Conflict, Diabetes Distress, Quality of Life, and Chronic Illness Care in Patients With Diabetes: Cluster Randomized Controlled Trial. *Journal of Medical Internet Research*. 2020;22(9):e16984.
49. Goelz T, Wuensch A, Stubenrauch S, Ihorst G, de Figueiredo M, Bertz H, et al. Specific training program improves oncologists' palliative care communication skills in a randomized controlled trial. *Journal of Clinical Oncology*. 2011;29(25):3402-7.
50. Schafer I, Kaduszkiewicz H, Mellert C, Loffler C, Mortsiefer A, Ernst A, et al. Narrative medicine-based intervention in primary care to reduce polypharmacy: results from the cluster-randomised controlled trial MultiCare AGENDA. *BMJ Open*. 2018;8(1):e017653.
51. Berendonk C, Kaspar R, Bär M, Hoben M. Improving Quality of Work life for Care Providers by Fostering the Emotional well-being of Persons with Dementia: A Cluster-randomized Trial of a Nursing Intervention in German long-term Care Settings *Dementia* 2019;18(4):1286-309.
52. Eggers C, Dano R, Schill J, Fink GR, Hellmich M, Timmermann L. Patient-centered integrated healthcare improves quality of life in Parkinson's disease patients: a randomized controlled trial. *J Neurol*. 2018;265(4):764-73.
53. Mills PD, Harvey PW. Beyond community-based diabetes management and the COAG coordinated care trial. *Australian Journal of Rural Health*. 2003;11(3):131-7.
54. Reed RL, Roeger L, Howard S, Oliver-Baxter JM, Battersby MW, Bond M, et al. A self-management support program for older Australians with multiple chronic conditions: A randomised controlled trial. *Medical Journal of Australia*. 2018;208(2):69-74.
55. Young JM, Butow PN, Walsh J, Durcinoska I, Dobbins TA, Rodwell L, et al. Multicenter randomized trial of centralized nurse-led telephone-based care coordination to improve outcomes after surgical resection for colorectal cancer: the CONNECT intervention. *Journal of Clinical Oncology*. 2013;31(28):3585-91.
56. Or C, Tao D. A 3-Month Randomized Controlled Pilot Trial of a Patient-Centered, Computer-Based Self-Monitoring System for the Care of Type 2 Diabetes Mellitus and Hypertension. *Journal of Medical Systems*. 2016;40(4):81.

57. Yu DSF. Effects of a Health and Social Collaborative Case Management Model on Health Outcomes of Family Caregivers of Frail Older Adults: Preliminary Data from a Pilot Randomized Controlled Trial. *Journal of the American Geriatrics Society*. 2016;64(10):2144-8.
58. Ko FWS, Cheung NK, Rainer TH, Lum C, Wong I, Hui DSC. Comprehensive care programme for patients with chronic obstructive pulmonary disease: A randomised controlled trial. *Thorax*. 2017;72(2):122-8.
59. Windrum P, Garcia-Goni M, Coad H. The Impact of Patient-Centered versus Didactic Education Programs in Chronic Patients by Severity: The Case of Type 2 Diabetes Mellitus. *Value in Health*. 2016;19(4):353-62.
60. Kennedy A, Nelson E, Reeves D, Richardson G, Roberts C, Robinson A, et al. A randomised controlled trial to assess the impact of a package comprising a patient-orientated, evidence-based self-help guidebook and patient-centred consultations on disease management and satisfaction in inflammatory bowel disease. *Health Technology Assessment*. 2003;7(28).
61. Kinmonth AL, Woodcock A, Griffin S, Spiegel N, Campbell MJ. Randomised controlled trial of patient centred care of diabetes in general practice: Impact on current wellbeing and future disease risk. *British Medical Journal*. 1998;317(7167):1202-8.
62. Alamo MM, Moral RR, Perula de Torres LA. Evaluation of a patient-centred approach in generalized musculoskeletal chronic pain/fibromyalgia patients in primary care. *Patient Education & Counseling*. 2002;48(1):23-31.
63. de Batlle J, Massip M, Vargiu E, Nadal N, Fuentes A, Ortega Bravo M, et al. Implementing Mobile Health-Enabled Integrated Care for Complex Chronic Patients: Intervention Effectiveness and Cost-Effectiveness Study. *JMIR Mhealth Uhealth*. 2021;9(1):e22135.
64. Machado LA, Azevedo DC, Capanema MB, Neto TN, Cerceau DM. Client-Centered Therapy vs Exercise Therapy for Chronic Low Back Pain: A Pilot Randomized Controlled Trial in Brazil. *Pain Medicine*. 2007;8(3):251-8.
65. Kikkenborg Berg S, Stoier L, Moons P, Zwisler AD, Winkel P, Ulrich Pedersen P. Emotions and health: findings from a randomized clinical trial on psychoeducational nursing to patients with implantable cardioverter defibrillator. *The Journal of cardiovascular nursing*. 2015;30(3):197-204.
66. Arian M, Memarian R, Oghazian MB, Vakilian F, Badiie Z. The effect of a holistic care program on the reduction of iron overload in patients with beta-thalassemia major: A randomized clinical trial. *Iranian Red Crescent Medical Journal*. 2018;20 (4) (no pagination)(e60820).
67. Lowther K, Selman L, Simms V, Gikaara N, Ahmed A, Ali Z, et al. Nurse-led palliative care for HIV-positive patients taking antiretroviral therapy in Kenya: a randomised controlled trial. *Lancet HIV*. 2015;2(8):e328-34.
68. Slok AH, Kotz D, van Breukelen G, Chavannes NH, Rutten-van Molken MP, Kerstjens HA, et al. Effectiveness of the Assessment of Burden of COPD (ABC) tool on health-related quality of life in patients with COPD: a cluster randomised controlled trial in primary and hospital care. *BMJ Open*. 2016;6(7):e011519.
69. Martin IR, McNamara D, Sutherland FR, Tilyard MW, Taylor DR. Care plans for acutely deteriorating COPD: a randomized controlled trial. *Chronic respiratory disease*. 2004;1(4):191-5.
70. Berntsen GKR, Dalbakk M, Hurley JS, Bergmo T, Solbakken B, Spansvoll L, et al. Person-centred, integrated and pro-active care for multi-morbid elderly with advanced care needs: a propensity score-matched controlled trial. *BMC health services research*. 2019;19(1):682-.
71. Low LL, Tan SY, Ng MJM, Tay WY, Ng LB, Balasubramaniam K, et al. Applying the integrated practice unit concept to a modified virtual ward model of care for patients at highest risk of readmission: A randomized controlled trial. *PLoS ONE*. 2017;12 (1) (no pagination)(e0168757).
72. Wichit N, Mnatzaganian G, Courtney M, Schulz P, Johnson M. Randomized controlled trial of a family-oriented self-management program to improve self-efficacy, glycemic control and quality of life among Thai individuals with Type 2 diabetes. *Diabetes Research and Clinical Practice*. 2017;123:37-48.
73. Farquhar MC, Prevost AT, McCrone P, Brafman-Price B, Bentley A, Higginson IJ, et al. Is a specialist breathlessness service more effective and cost-effective for patients with advanced cancer and their carers than standard care? Findings of a mixed-method randomised controlled trial. *BMC Medicine*. 2014;12(1):194.
74. Farquhar MC, Prevost AT, McCrone P, Brafman-Price B, Bentley A, Higginson IJ, et al. The clinical and cost effectiveness of a Breathlessness Intervention Service for patients with advanced non-malignant disease and their informal carers: mixed findings of a mixed method randomised controlled trial. *Trials*. 2016;17(1):185.

VERSION 2 – REVIEW

REVIEWER	Senjam, Suraj Singh All India Institute of Medical Sciences, Community Ophthalmology, Dr. Rajendra Prasad Centre for Ophthalmic Sciences
REVIEW RETURNED	16-May-2022
GENERAL COMMENTS	Thanks to all authors for the revision There so many 'n', mentioning in the entire manuscript. It can be reduced comfortably at some places.
REVIEWER	Gyllensten, Hanna University of Gothenburg, Institute of Health and Care Sciences
REVIEW RETURNED	15-Jun-2022
GENERAL COMMENTS	Thank you for your careful revisions. All my questions and concerns have been well handled.